# Fundamental nursing care focusing on older people's needs and continuity of long-term care: a scoping review protocol

Ole Martin Nordaunet [1,2] Edith Roth Gjevjon [2] Cecilia Olsson [1]
Hanne Aagaard [3] Gunilla Borglin [2]

¹Institute of Health Sciences, Department of Nursing, Faculty of Health Science and Technology, Karlstad University, Karlstad, Sweden
²Bachelor of Education in Nursing, Lovisenberg Diaconal University College, Oslo, Norway
³Lovisenberg Diaconal University College, Oslo, Norway

**Correspondence to**
Ole Martin Nordaunet;
ole.martin.nordaunet@ldh.no

## ABSTRACT

**Introduction** Knowledge about long-term care services ability, regardless of if the service is home-based or facility-based, to provide an optimal and comprehensive fundamental nursing care (understood as focusing on physical, relational and psychosocial needs) consistently over time is sparse. Research into nursing indicates the presence of a discontinuous and fragmented healthcare service, and that fundamental nursing care such as mobilisation, nutrition and hygiene among older people (65 years and above) seems to be, regardless of reasons, systematically rationed by nursing staff. Thus, our scoping review aims to explore the published scientific literature on fundamental nursing care and continuity of care targeting older people's needs while also describing identified nursing interventions with the same foci in a long-term care context.

**Methods and analysis** The upcoming scoping review will be conducted in accordance with Arksey and O'Malley's methodological framework for scoping studies. Search strategies will be developed and adjusted to each database, for example, PubMed, CINAHL and PsychINFO. Searches will be limited to the years 2002–2023. Studies focusing our aim, regardless of study design, will be eligible for inclusion. Included studies will be quality assessed and data will be charted using an extraction form. Textual data will be presented through a thematic analysis and numerical data by a descriptive numerical analysis. This protocol adheres to the Preferred Reporting Items for Systematic Reviews and Meta-Analyses protocol checklist.

**Ethics and dissemination** The upcoming scoping review will take into consideration ethical reporting in primary research as part of the quality assessment. The findings will be submitted to an open-access peer-reviewed journal. Under the Norwegian Act on Medical and Health-related Research, this study does not need ethical clearance by a regional ethical review authority as it will not generate any primary data or obtain sensitive data or biological samples.

## INTRODUCTION

As the older part of the population is increasing and living longer, more and more older people find themselves in

**STRENGTHS AND LIMITATION OF THIS STUDY**

⇒ The combination of quality assessment, reporting guidelines and a well-established methodology will support 'clear statements about where the specific focus should be and importantly, what does not require further research.

⇒ Including qualitative, quantitative and mixed methods design will assist us to achieve a comprehensive picture of the present knowledge base.

⇒ Excluding grey literature might limit our results.

need of long-term care.[1] Research implies that the demand for long-term care will rise substantially all over Europe.[2] Nowadays, older people are also more often described to have complex care needs mainly due to old age and multiple chronic conditions. Consequently, they are correspondingly described as requiring comprehensive fundamental nursing care, related to nutrition, mobility, elimination, personal hygiene, etc.[3–5] Attending to such needs lies well within the remit of nursing. Particularly as we, like Henderson,[6] recognise fundamental nursing care to be directed at assisting with a person's personal needs such as nutrition, cleanliness, elimination, mobility, etc. Thus, 'assisting people to do things they would normally do for themselves if only they were able' (p149).[7] Fundamental nursing care is described as combining physical, relational and psychosocial dimensions of nursing care encounters.[8 9] Research[10 11] has described fundamental nursing care to be in a poor condition, resulting in findings unfit to provide any evidence-based guidance. In-depth knowledge about the scope and practice of fundamental nursing care and of the practice setting is therefore of vital

importance to support nursing staff's skills and abilities in delivering optimal, fundamental nursing care to older people in need of help and support.

Most of the workforce responsible to deliver fundamental nursing care in the long-term care context, regardless of the service, is home-based or facility-based, consists of nursing staff (registered, licensed, diploma nurses and nurse assistants). Some literature reviews highlight that the nursing staff within this increasingly highly specialised context need to represent a wide variety of competencies.[12 13] Knowledge about long-term care services' ability to provide optimal and comprehensive care targeting fundamental nursing care (understood as physical, relational and psychosocial needs) consistently over time is sparse.[14 15] Continuous care (ie, continuity of care) is often described as a key indicator of the concept of quality of care. As a key indicator, it is known to prevent unnecessary hospital admissions, lowering cost as well as decreasing mortality.[16–18] Ensuring continuity of care impacts the quality of care and vice versa. Moreover, research into nursing suggests the presence of a discontinuous and fragmented healthcare service, and that needs such as mobilisation, nutrition and hygiene among older people are likely to become rationed by nursing staff.[19 20] Thus, unmet or rationed care is common in the long-term care context.[20 21]

It has been described that nursing staff working in accordance with distinctly articulated nursing interventions such as different, well-defined models of care, patient care pathways or in accordance with clinical practice guidelines are more likely to work systematically and towards mutually set goals in care.[3] Hence, the deliverance of care can more easily be assessed and evaluated on a regular basis. Implementing such strategies to care which includes structured assessments and evaluations might be one way to aid nursing staff to deliver fundamental nursing care in a continuous and optimal way. It could also promote fundamental nursing care in a more predictable manner that corresponds to older patients' preferences and needs continuously over time. However, recent literature reviews suggest that the implementation of distinctly articulated and well-defined nursing interventions (understood as the aforementioned description) targeting fundamental nursing care and/or continuity of care among older people in the long term care is scant.[22 23] Consequently, the development and testing of viable nursing interventions supporting nursing staff skills and abilities in relation to the deliverance of a continuously and optimally fundamental nursing care among older people are vital in the long-term care context to ensure both continuity and quality of care. Accordingly, our scoping review aims to explore the published scientific literature on fundamental nursing care and continuity of care targeting older people's needs while also describing identified nursing interventions with the same foci (box 1) in a long-term care context.

---

**Box 1  Operationalisation of tentative core concepts in the review**

⇒ Long-term care is operationalised here as healthcare delivered over prolonged periods of time in the patient's home, nursing home or sheltered housing but can as well be delivered in other contexts in the community, such as, but not limited to, day care centres, assisted living facilities and retirement homes.[1 54–56]

⇒ Fundamental nursing care is operationalised here as care directed at assisting with a person's personal needs such as mobility, nutrition, cleanliness, etc, combined with physical, psychosocial and relational dimensions.[8 9]

⇒ Continuity of care is operationalised here as 'The degree to which a series of discrete health care events is experienced by people as coherent and interconnected over time and consistent with their health needs and preferences' (p8).[57]

⇒ Key stakeholders are operationalised here as older people and nurses.

⇒ Nurses and nursing staff are operationalised here as practising nurses, professional nurses, associate professional nurses, healthcare assistants, nursing aides, nurse specialists, registered nurses, registered practical and licensed practice nurses, hereafter referred to as nurses.[58 59]

⇒ Nursing interventions are operationalised here as distinctly articulated and defined nursing activities, models of care, patient care pathways or clinical practice guidelines (cf Davidson et al[60]).

---

## METHODS

Our scoping review will be based on Arksey and O'Malley's[24] methodological framework as well as on updated methodology.[25–27] Scoping review methodology is particularly useful in assessing key characteristics and identifying knowledge gaps in the literature within complex topics.[24 28 29] Two authors (ERG and GB) will collaborate closely with the first author (OMN) during the process, while two authors (HA and CO) will provide critical assessment and feedback within the overall process of stages. This protocol is designed and reported in accordance with the Preferred Reporting Items for Systematic Review and Meta-Analysis Protocols checklist (online supplemental material 1).[30] Our upcoming scoping review will adhere to the Preferred Reporting Items for Systematic Reviews and Meta-Analyses (PRISMA)-Scoping Review checklist[31] and is registered at Open Science Framework, last updated on 9 March 2023 (https://doi.org/10.17605/OSF.IO/XJ39E).

### Stage 1: identifying the research questions

The tentative questions to the literature for our upcoming scoping review are deliberately broad to cover the breadth of research evidence in our field of foci. Thus, we are aiming for comprehensiveness. One of the strengths with scoping reviews is the iterative methodological process which will allow us to decide to add supplementary questions or to fine-tune our suggested questions based on the findings emerging during the review process.[24] A modified population, intervention, comparison, outcome and study setting (PICOS) will support us in our scoping

**Table 1** Framework for determining eligibility of the tentative research questions

| Criteria | Determinants |
|---|---|
| Population | Older people (65 years and above), nurses (box 1) and relatives in a broad understanding |
| Intervention (phenomenon of interest) | Fundamental nursing care and nursing interventions (box 1) targeting older people's needs |
| Comparison | N/A |
| Outcome | Physical, relational or psychosocial needs Continuity of care Nursing interventions targeting continuity and/or fundamental nursing care |
| Study setting | Home-based or facility based long-term care (box 1) for older people |

NA, not applicable.

review (table 1) and in deciding the suitability of our tentative questions to the literature.[32 33]

Our upcoming scoping review will tentatively explore the following questions to the literature:

► What fundamental nursing care (box 1) are described in the literature as targeting older people's need in a long-term care context?
► How is fundamental nursing care described and experienced by key stake holders (box 1) in a long-term care context?
► What fundamental nursing care interventions (box 1) are described in the literature targeting older people's fundamental needs or their continuity of care in a long-term care context?

Additionally, subquestions will be used to facilitate an in-depth exploration of the included papers and their findings. Subquestions will tentatively encompass the following: what characterises the long-term care settings described? who's perspective is investigated (eg, stakeholders), type of nurses delivering fundamental nursing care? which outcomes are in focus and country of origin.

## Stage 2: identifying relevant studies

Searches will be conducted in the three databases, PubMed, CINAHL and PsycINFO, covering the majority of published peer-reviewed health service research. The search strategies will be developed following PICOS (table 1). The first author consulted an information specialist at Karlstad University, Sweden, to refine the initial tentative search strategy (online supplemental file 2). As part of the iterative process, our search strategies will be further tested and refined before executing the upcoming scoping review. The final search strategies for each individual database will also be reviewed by an information specialist at Karlstad University, Sweden. To reflect the iterative approach of scoping reviews, it is important to strike a balance between comprehensiveness and manageability of search results.[26] Accordingly, we will

include controlled subject headings (Medical Subject Headings), keywords, synonyms and Boolean operators.[34] The searches will be time limited to 2002–2023. We chose to begin our searches in year 2002 as initial test searchers indicated a lack of published peer-reviewed papers in relation to fundamental nursing care before this year. Included papers' reference lists will be searched.[34 35] Considering what types of research likely to answer our tentative questions to the literature, but also resources and time needed to scan for grey literature,[35] We will only include published peer-reviewed research in English.

## Stage 3: study selection

When searches have been applied in the databases, eligible papers will be imported to EndNote[36] giving us a clear overview of the net selection while also removing any duplicates. The remaining papers will then be imported into Rayaan.[37] The first (OMN) and last author (GB) will review 5%–10% of the gross selection, ensuring an agreed approach for deciding ineligible and eligible papers. To help guide the title and abstract screening, the PICOS framework (table 1) will be used to support the process (cf Polanin *et al*)[38] mainly as the eligible criteria relate to our determinants in the framework. For example, all study designs are eligible, and our population will be limited to nurses (box 1) and older people defined as 65 years and older. Our rationale for the latter is that 65 years is one of the standard cut-offs for older people in most research and databases. However, considering the iterative process of a scoping review, the eligibility criteria are likely to be further developed as our knowledge in the field accumulate.[24] The Rayyan interface will additionally use the PICOS framework (table 1) to further support us in the screening process. Research type, language and ethical considerations will also be part of our eligible criteria. During the title and abstract screening, we plan to have a 'sifting' approach, narrowing down the net selection while resolving conflicts in Rayyan.[37] Frequent meetings are planned to support us in the sifting approach. This strategy has been described as a successful strategy by others.[39] On completion of the study selection, the research team will evaluate the eligible papers' suitability before the assessment and data charting will begin. The process as whole will be recorded with a logbook and the search process will be illustrated using the PRISMA flowchart.[40]

## Stage 4: data charting

Eligible papers subject to full-text assessment will be summarised in an extraction chart, and the tentatively planned extraction data can be viewed in box 2. Additionally, the assessment of methodological quality using critical assessment tools will aid us in identifying papers with poor methodological rigour. Identification of such papers may suggest that any results and findings are based on research with improper declaration of research methodology; as such, any generalisable results will be used cautiously. Although a quality assessment was not

---

> **Box 2 Tentative data for extraction chart**
>
> Data
> ⇒ ID number.
> ⇒ Author.
> ⇒ Year.
> ⇒ Country of origin.
> ⇒ Aim, objective and/or research questions.
> ⇒ Study context and description/definition of long-term care.
> ⇒ Key stakeholder (who).
> ⇒ How is 'nurse' described/defined.
> ⇒ Research design.
> ⇒ Sample (who and how many).
> ⇒ Data collection.
> ⇒ Data analysis.
> ⇒ Summary of findings.
> ⇒ Descriptions of components in nursing interventions and outcomes.
> ⇒ Quality appraisal adjusted to research design.
> ⇒ Description of ethics.[41]
> ⇒ Approved by a research ethical committee?
> ⇒ Was informed consent obtained?
> ⇒ Adequate data protection?
> ⇒ Declaration of financial support?
> ⇒ Declaration on potential conflicts of interest?

initially recommended by the originators, later developments have seen quality assessment as a necessary component.[24 26 28] Critical appraisal, that is, quality assessment of included papers, will be done in accordance with their study design. We will implement Weingarten et al[41] ethical assessment for systematic reviews. The first, second and last authors will pilot 20% of the eligible papers to 'ensure that all necessary data is captured appropriately' (p6),[27] the data extraction chart will be revised after the piloting if needed. Eligible papers with questionable method or ethical reporting will be discussed by the team.

### Stage 5: collating, summarising and reporting results

Arksey and O'Malley[24] highlight that the analytical framework used within scoping reviews should support an overview and presentation of a potentially large body of material (p28). A basic numerical analysis, mapping charts and tables in combination with a thematic analysis is common when collating, summarising and reporting results within scoping reviews. Both Levac et al[28] and Daudt et al[26] state that applying meaning to the results is important, and especially understanding the implications within a broader setting which can include policy, contextual and practice implications. However, scoping reviews have been contested as failing to provide an actual scope of the research literature, and the high rejection rate among scoping reviews can be pinpointed to poor reporting, superficial presentation of results and lack of methodological guidance in this stage.[42–44] One effort to mitigate some of these issues was proposed by Bradbury-Jones et al,[25] further developing the reporting of results through adding clarity and focus on reporting Patterns, Advances, Gaps, Evidence for Practice and Research Recommendations, presented as the PAGER framework.

This analytical approach can aid the research team by offering structure, identify interconnections, identify advances and gaps, and support reporting.[25] Besides using the PAGER framework, our analytical process will be aided by the recommendation of Levac et al,[28] suggesting a qualitative thematic analysis,[45] focusing on semantic and manifest descriptions to summarise the content as well as basic descriptive presentation of quantitative data[24] and visual displays.[27 42] The research team will hold regular meetings discussing results using key reflective questions as proposed by Bradbury-Jones et al.[25]

### Patient and public involvement

Patients and/or the public were not involved in the design, conduct, reporting, or dissemination plans for this research.

## ETHICS AND DISSEMINATION

Under the Norwegian Act on Medical and Health-related Research,[46] the upcoming scoping review does not need ethical clearance by a regional ethical review authority as it will not generate any primary data or obtain sensitive data or biological samples. It is still important to note that, for example, Weingarten et al[41] advocated early on for the need for ethical assessment in literature reviews. This was later concurred by Wager and Wiffen,[47] who states that researchers conducting literature reviews have an ethical responsibility, being transparent, accurate, identifying and reporting fraudulent research. These research principles align with established ethical principles, namely, reliability, honesty, respect, and accountability.[48] The planning of this scoping review is developed in unity with recommendations on ensuring value in research, with particular emphasis on appropriate research design, development of protocol and justifiable research priorities.[49]

The results from the upcoming scoping review will support us in the initial phase of the Medical Research Council framework for complex interventions.[50] Thus, the knowledge gained from the upcoming scoping review can serve as a theoretical underpinning for the possible development of an educational intervention targeting fundamental nursing care aiming at continuity of care and older people's needs in a long-term care setting.

In the upcoming scoping review, we will implement reporting guidelines,[30 31] conduct quality assessments of included papers method and ethical reporting, adhere to updated guidelines on the scoping review methodology[25 27] and use a designated protocol.[51] Thus, it is likely that the upcoming scoping review will support us in making 'clear statements about where the specific focus should be and importantly, what does not require further research' (p68).[52] Implementing such strategies can also help to reduce the probability of producing research waste.[53]

**Acknowledgements** Our sincere gratitude goes to information specialists Annelie Ekberg-Andersen and Linda Borg at Karlstad University, Sweden, for their support in the development of our initial search strategies.

**Contributors** OMN wrote the initial draft. GB, ERG, CO and HA contributed with important intellectual content and critical revision. OMN and GB developed the search strategy, and GB supervised the study. All authors were responsible for the study's inception and design and read and approved the final manuscript.

**Funding** The authors have not declared a specific grant for this research from any funding agency in the public, commercial or not-for-profit sectors.

**Competing interests** None declared.

**Patient and public involvement** Patients and/or the public were not involved in the design, conduct, reporting or dissemination plans of this research.

**Patient consent for publication** Not applicable.

**Provenance and peer review** Not commissioned; externally peer reviewed.

**ORCID iDs**
Ole Martin Nordaunet http://orcid.org/0000-0002-6061-0033
Edith Roth Gjevjon http://orcid.org/0000-0002-9656-522X
Cecilia Olsson http://orcid.org/0000-0002-0944-5650
Hanne Aagaard http://orcid.org/0000-0002-7810-9298
Gunilla Borglin http://orcid.org/0000-0002-7934-6949

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
