## [Reviewer comments · BMJ Open]

ARTICLE DETAILS

TITLE (PROVISIONAL)	Fundamental Nursing Care Focusing Older Peoples Needs and Continuity of Long-Term Care: A Scoping Review Protocol
AUTHORS	Nordaunet, Ole Martin; Gjevjon, Edith; Olsson, Cecilia; Aagaard, Hanne; Borglin, Gunilla

VERSION 1 – REVIEW

REVIEWER	Jones, K The Open University, HWSC WELS
REVIEW RETURNED	15-Nov-2022

GENERAL COMMENTS	This is a welcome scoping review that you intend to undertake. You have provided a thorough account about how this will be achieved. The methodological approach is sound and appropriate for the intended scope of the literature pertaining to nursign and care models pertaining to needs of older people.
---

REVIEWER	Giosa, Justine Saint Elizabeth Health Care
REVIEW RETURNED	10-Feb-2023

GENERAL COMMENTS	Thank you for the opportunity to review your scientific work. The topic of long-term care, nursing and meeting older adults' needs is very timely. Overall, I found the methods of your paper to be transparent in terms of the steps taken for the scoping review itself (with a few questions- see below). However, I found the context, framing, objectives, research questions and their link to the outcomes to be too broad/ vague and a bit to complex in number of concepts being linked. I hope you find the comments below helpful. Title  -The title is a bit hard to follow/ has a lot of concepts. You are talking about 1) nursing care 2) models of care (are these related?) and 3) older adults; 4) basic care needs; 5) continuity and 6) LTC and then a scoping review. -would advise a more streamlined/ clear title -is there a particular geography of focus for the work? If so, I recommend to add it in as BMJ open is an international audience Abstract  -It is unclear as the reader what is meant by 'long-term care in the community' - LTC looks very different around the world -'basic needs' and 'older people' are also undefined and could vary in their definition(s) -Re: the overall aim statement: 'aim to map the literature on nursing care in relation to basic care, continuity of care and described nursing interventions targeting basic care among older adults in long-term care'. There are many concepts here, with massive bodies
--

of literature and it is not apparent to the reader what the overall goal of the work is from this statement. What does 'map' mean, and is that the objective of a scoping review?

-a stronger positioning on the rationale of the work, the specific literature gap and problem that is being addressed by the review is warranted

Strengths and limitations

-in this section, as a reader I am looking to be convinced as to what this paper adds net new to the published body of literature on this topic/ method and the bullet points and the bullet points here could be strengthened/ clarified to this end. For example, what does 'comprehensive development of search strategies for optimal uptake of papers' mean? Using good/ well-established methods/ quality assessment is an expectation of published papers, not a unique strength.

Introduction

-This section of the paper is the most challenging to interpret/ understand as the reader because there seems to be context missing in terms of how concepts are being defined and used AND/OR concepts are being positioned as too broad to hold specific and applicable meaning. Examples are below

-LTC is being positioned as an aging in place strategy but the definition posed is a Canadian definition--and in Canada, LTC is mostly a facility-based care concept despite this definition--aging in place would be facilitated more by home and community care and community support services sectors

-It would be helpful if you rooted your discussion about LTC in a particular geography—country/region/ etc. as LTC looks very different around the world

-No reference was provided to support the statement that people are spending less time in hospital-based care and instead in long-term care? I am not sure that is true, and depends on country/jurisdiction etc. and LTC is in many places still facility-based

-It is difficult to consider all 'long-term care contexts' as being the same--if you are including both facility-based and home care contexts, these are so variable that it would be hard to make relevant/ meaningful comparisons between care models in these settings.

-It is also unclear what the rationale is for looking at nursing models of care. For example, in facility-based LTC in Canada there are very few nurses, and most care is provided by unregulated PSWs and who are co-located in a building. More nurses work in the community, but in silos from other providers, and even still, unregulated (personal support workers or health care aides) provide the majority of care. Perhaps nurses provide the majority of LTC in other countries/ jurisdictions but a rationale / clarity is warranted. Also, some of the basic needs noted (e.g., function) would be guided by other allied health professionals (e.g., OT/PT/SLP/Social work). Why have they been excluded in thinking about meeting basic needs?

- It appears the main rationale for the work is that unmet basic needs in LTC are due to poor nursing care continuity. Again, this is context specific and see above comments re: setting/ other team members. Further, other factors in the system can lead to poor continuity and unmet needs (e.g, HHR challenges, fee-for-service payment structures etc.). These are not identified or discussed in your paper and it feels like there is too much onus being placed on the individual providers re: their autonomy to meet needs at the point-of-care and their 'poor understanding of basic needs of older people'- it

	almost sounds like providers are to blame, when we know it is often a systems issue, despite providers doing their best within the system  -The definition provided of what is meant by 'model of care' is vague -positioned as being a 'description'--from an implementation science perspective a description would not be actionable in the way you suggest for practice guidance in any long-term care context Methods  -Overall, there seems to be good alignment with Arksey & O'Malley's framework, and quality appraisal addition. PRISMA, PAGER considerations included, which is well thought out. -Family/ relatives are introduced as a concept in the methods but not discussed in the background/ rationale and should be if they are to be considered—there is a body of literature on family and family caregivers re: long-term care -While it is understandable for scoping review research questions to be broad, it is unclear how question 1 will address the research aim/ be answerable by a search strategy? -What is the rationale for older age being defined as 65+? -Why is literature inclusive only until 2022? -What will the selection criteria be for the title/ abstract screening phase? -The paper provides no indication as to whether the study is exploring literature from countries with similar LTC contexts? See earlier comments -In the methods there is mention that the scoping review is part of a larger study in Norway and Sweden- it would be helpful to situate the overall context of LTC in Norway and Sweden in background/ intro/ rationale etc. if that is where/ how the results will be applied/ used to inform the broader work -The description of the consultation phase of the scoping review is vague- how will the consultations inform the review itself? Who is actually part of the PPPI group? -No clear statement about ethics review (not needed, but should state why) Supplemental Files  -preliminary search strategy is very wide, which is aligned with the undefined nature of some of the concepts above—consider further honing
--	---

VERSION 1 – AUTHOR RESPONSE

Reviewer 1, Dr Jones

Thank you for your kind words regarding our manuscript and for taking the time to critically review it.

Reviewer 2, Dr Giosa

Thank you for the opportunity to review your scientific work. The topic of long-term care, nursing and meeting older adults' needs is very timely. Overall, I found the methods of your paper to be transparent in terms of the steps taken for the scoping review itself (with a few questions- see below). However, I found the context, framing, objectives, research questions and their link to the outcomes to be too broad/ vague and a bit too complex in number of concepts being linked. I hope you find the comments below helpful.

- Dear Dr. Giosa, thank you very much for taking the time to review our brief protocol for an upcoming scoping review, and for your valuable comments aiming at supporting the text to become more logical and succinct. Before addressing all your important concerns below, we just wish to stress that our choice of design is based upon the fact that a scoping review allows for initial broad questions to the literature as opposed to a systematic review. The scoping review design expects the researchers to refine, reformulate both searches and the questions to the literature in a constant ongoing iterative process as the work proceeds. Our choice of design was done because neither the setting (long-term care) or the concept of nurses is universally understood or described in the peer-reviewed nursing literature. Thus, working through the methodological process of searches, search blocks will support us to end up with more broad but narrow questions to the literature, and search blocks being more on point while informing us about where the strengths and the weaknesses are in our tentative focus for our upcoming review. Regardless of this we have done our utmost best to address your concerns and hope that this will eliminate most of your major concerns?

1#. The title is a bit hard to follow/ has a lot of concepts. You are talking about 1) nursing care 2) models of care (are these related?) and 3) older adults; 4) basic care needs; 5) continuity and 6) LTC and then a scoping review. would advise a more streamlined/ clear title

- Thank you for your comment, revised and we sincerely hope that it now has become clearer?

2#. Is there a particular geography of focus for the work? If so, I recommend adding it in as BMJ open is an international audience

- No, however we acknowledge the importance of geography. We aim to control for this component via our data extraction sheet for the upcoming review as well as via our tentative sub-questions to the literature. (Please see revised table 2 and section regarding tentative sub-question in manuscript)

3#. Abstract: It is unclear as the reader what is meant by 'long-term care in the community' - LTC looks very different around the world

-'basic needs' and 'older people' are also undefined and could vary in their definition(s)

-Re: the overall aim statement: 'aim to map the literature on nursing care in relation to basic care, continuity of care and described nursing interventions targeting basic care among older adults in long-term care'. There are many concepts here, with massive bodies of literature and it is not apparent to the reader what the overall goal of the work is from this statement. What does 'map' mean, and is that the objective of a scoping review?

-a stronger positioning on the rationale of the work, the specific literature gap and problem that is being addressed by the review is warranted

- Abstract have been carefully gone through and revised to meet issues raised above as have main body of text to reflect our revised abstract. Thank you for drawing our attention to the concept of basic needs which is now revised to better reflect our intentions which is fundamental nursing care targeting older peoples needs. We sincerely hope that the revised introduction will support the reader to understand the main concepts in this protocol – all vital concepts are additionally operationalised in box 1
- Aim of study revised as is the tentative questions to the literature

#4. Strengths and limitations

- In this section, as a reader I am looking to be convinced as to what this paper adds net new to the published body of literature on this topic/ method and the bullet points, and the bullet points here

could be strengthened/ clarified to this end. For example, what does 'comprehensive development of search strategies for optimal uptake of papers' mean? Using good/ well-established methods/ quality assessment is an expectation of published papers, not a unique strength.

- Thank you – this section is carefully revised in accordance with the above and changes highlighted in yellow

5#. Introduction: This section of the paper is the most challenging to interpret/ understand as the reader because there seems to be context missing in terms of how concepts are being defined and used AND/OR concepts are being positioned as too broad to hold specific and applicable meaning. Examples are below. LTC is being positioned as an aging in place strategy, but the definition posed is a Canadian definition--and in Canada, LTC is mostly a facility-based care concept despite this definition--aging in place would be facilitated more by home and community care and community support services sectors.

It would be helpful if you rooted your discussion about LTC in a particular geography—country/region/ etc. as LTC looks very different around the world.

- Thank you for sharing your concerns regarding the concept of long-term care. Regardless of our chosen definitions origin it encompasses the upcoming scoping reviews focus, that is to include studies conducted both in home-based long-term care and in facility based long-term care additionally reflected in our very tentative search blocks. Thus, the team is aware of the above, no universal understanding of LTC, as well as of the importance to not ending up comparing apple with pears. Therefore, one of the sub-questions in this scoping review are focusing on the included papers description of setting/context consequently data will be extracted in this regard for subsequent descriptive analysis. Additionally, data on country of study execution will be extracted thus limiting our scoping review to a particular geography will limit our possibilities to explore the concept of long-term care from an international perspective. We sincerely hope that our assurance of keeping context/setting, type of health care professionals, country in control for the upcoming scoping review will be acceptable?
- Additionally, the introduction has been revised, hopefully in accordance with your reflections/comments above and we hope that our revision now might have eliminated your concerns regarding the above comment.

6#. No reference was provided to support the statement that people are spending less time in hospital-based care and instead in long-term care? I am not sure that is true and depends on country/ jurisdiction etc. and LTC is in many places still facility based.

- Introduction revised.

7#. It is difficult to consider all 'long-term care contexts' as being the same--if you are including both facility-based and home care contexts, these are so variable that it would be hard to make relevant/ meaningful comparisons between care models in these settings.

- Revised please see our answer in 2#/5#

8#. *It is also unclear what the rationale is for looking at nursing models of care. For example, in facility based LTC in Canada there are very few nurses, and most care is provided by unregulated PSWs and who are co-located in a building. More nurses work in the community, but in silos from other providers, and even still, unregulated (personal support workers or health care aides) provide most of the care. Perhaps nurses provide the majority of LTC in other countries/ jurisdictions, but a rationale / clarity is warranted. Also, some of the basic needs noted (e.g., function) would be guided by other allied health professionals (e.g., OT/PT/SLP/Social work). Why have they been excluded in thinking about meeting basic needs?*

- Thank you for drawing our attention to your above concern. As with the concept of long-term care the concept of nurse is equally muddled in the literature. Nurses in our scoping review are broadly operationalised (Box 1) as; Practicing nurses, professional nurses, associate professional nurses, healthcare assistants, nursing aides, nurse specialists, registered nurses, registered practical, licenced practice nurses. As the case with context/setting (facility based long-term care or home-based long-term care) data will be extracted concerning type of nurse professionals. Our focus on fundamental nursing care and older people's needs in this context (which also will be extracted as part of our findings) can certainly be recognised to be carried out by non-registered 'assistants', however we conceive of these assistants as the operational arm of registered nurses (RNs), carrying out nursing behaviours under supervision and leadership from RNs. Thus, as you yourself highlights in your last comment "preliminary searches are very wide" it would not be feasible to further broaden them outside the operationalisation of "nursing, nurses and nursing care" to include allied health care professionals.
- Regarding your concern about models of care the introduction have been carefully gone through and revised including the paragraph about MOC. Thank you for your help to further operationalise and define this concept in introduction and in box 1 (Under the heading methods). Our focus is nursing interventions defined as distinctly articulated and defined nursing activities which can include MOC, patient care pathways and clinical practice guidelines which can be viewed as working in an evidence-based manner.

9#. *It appears the main rationale for the work is that unmet basic needs in LTC are due to poor nursing care continuity. Again, this is context specific and see above comments re: setting/ other team members. Further, other factors in the system can lead to poor continuity and unmet needs (e.g, HHR challenges, fee-for-service payment structures etc.). These are not identified or discussed in your paper, and it feels like there is too much onus being placed on the individual providers re: their autonomy to meet needs at the point-of-care and their 'poor understanding of basic needs of older people'- it almost sounds like providers are to blame, when we know it is often a systems issue, despite providers doing their best within the system*

- Thank you for addressing the above it has hugely helped us to revise and hopefully present our intentions/rationale/reasons in a clearer more succinct manner. The aim of the study is revised and so are our tentative questions to the literature – we realized that we had not managed to convey our intent at all. We aim to explore the literature on fundamental nursing care and continuity of care while also describing identified nursing interventions (operationalised and defined in Box 1 method) targeting older people's needs in long-term care. Thus, exploring the present knowledgebase in LTC re fundamental nursing care, older peoples described needs in this setting and if fundamental nursing care are described in organised, systematic structured ways through nursing iv, MOCs, PCP, or CPG and if the concept of continuity are described as component or part of FNC or these IV. The brief introduction is carefully gone through and revised when deemed of importance, thank you again. We sincerely hope that our revisions can remove the concerns described above.

10#. *The definition provided of what is meant by 'model of care' is vague –positioned as being a 'description'--from an implementation science perspective a description would not be actionable in the way you suggest for practice guidance in any long-term care context.*

- Introduction and this paragraph are carefully revised and partially addressed by us in our replies 8# and 9#

11#. *Methods: Overall, there seems to be good alignment with Arksey & O'Malley's framework, and quality appraisal addition. PRISMA, PAGER considerations included, which is well thought out. Family/ relatives are introduced as a concept in the methods but not discussed in the background/ rationale and should be if they are to be considered—there is a body of literature on family and family caregivers re: long-term care.*

- Thank you for drawing our attention to this – to narrow our search strategies the research team has decided to only focus on the older persons and nurses' perspective in the upcoming scoping review.

12#. *While it is understandable for scoping review research questions to be broad, it is unclear how question 1 will address the research aim/ be answerable by a search strategy?*

- We are not sure we are able to follow the above (then sorry), our initial test searches both conducted alone and with an informational specialist indicates no issue. We do however acknowledge that it could have been an issue with the originally badly formulated aim and tentative questions to the literature – but we hope our carefully revision throughout this protocol now have eliminated this issue.

9#. *What is the rationale for older age being defined as 65+?*

- Older people will be defined as all people over the age of 65 years, considering that this is a standard cut-off for older people in most research and databases today. This is clarified in the text.

10#. *Why is literature inclusive only until 2022?*

- 2022 was the year this manuscript was submitted – year range now updated.

11#. *What will the selection criteria be for the title/ abstract screening phase?*

- Revised under the heading study selection.

12#. *The paper provides no indication as to whether the study is exploring literature from countries with similar LTC contexts? See earlier comments.*

- Our literature review will include any international published peer-reviewed papers meeting our inclusion criteria's regardless of country – we hope that our earlier replies #2 and #5 has eliminated this issue?

13#. *In the methods there is mention that the scoping review is part of a larger study in Norway and Sweden- it would be helpful to situate the overall context of LTC in Norway and Sweden in background/ intro/ rationale etc. if that is where/ how the results will be applied/ used to inform the broader work*

- This is a brief protocol for an upcoming literature review where the findings coming out of the review aims to support the research team to understand what to particularly focus on in the next steps, what could be of importance regarding choice of our setting/context, which type of nursing professionals should be targeted and how and/or what type of fundamental nursing care needs would be of importance to focus among older people, are there any nursing interventions (see operationalization box 1) described that can be modified, implemented etc. We hope the careful revision of the whole manuscript can be evaluated as acceptable. The larger study departures from the UK MRC-framework in which reviews are viewed as vital to iteratively support the design and research focus of the coming studies.

14#. *The description of the consultation phase of the scoping review is vague- how will the consultations inform the review itself? Who is part of the PPPI group?*

- Revised this step of the process have been removed as it is not a mandatory step when not including other literature than published peer-reviewed papers. Thank you for drawing our attention to this.

15#. *No clear statement about ethics review (not needed, but should state why)*

- Revised in abstract and last in manuscript

16#. *Supplemental Files preliminary search strategy is very wide, which is aligned with the undefined nature of some of the concepts above—consider further honing.*

- We hope that our overall revision of this study protocol and explanations in the method section will be acceptable – this is an early tentative search strategy (see revision under the heading identifying relevant studies.

Once again thank you for your engagement and valuable feed-back on our text – we hope our revision would be deemed acceptable.